# Task-Agnostic Low-Rank Adapters for Unseen English Dialects

**Zedian Xiao**🌲 **William Held**🐝 **Yanchen Liu**🛡️ **Diyi Yang**🌲

Stanford University🌲 Georgia Institute of Technology🐝 Harvard University🛡️
markxiao@stanford.edu, wheld3@gatech.edu, yanchenliu@g.harvard.edu,
diyiy@cs.stanford.edu

## Abstract

Large Language Models (LLMs) are trained on corpora disproportionally weighted in favor of Standard American English. As a result, speakers of other dialects experience significantly more failures when interacting with these technologies. In practice, these speakers often accommodate their speech to be better understood. Our work shares the belief that language technologies should be designed to accommodate the diversity in English dialects and not the other way around. However, prior works on dialect struggle with generalizing to evolving and emerging dialects in a scalable manner. To fill this gap, our method, **Hyper-LoRA**, leverages expert linguistic knowledge to enable resource-efficient adaptation via hypernetworks. By disentangling dialect-specific and cross-dialectal information, HyperLoRA improves generalization to unseen dialects in a task-agnostic fashion. Not only is HyperLoRA more scalable in the number of parameters, but it also achieves the best or most competitive performance across 5 dialects in a zero-shot setting. In this way, our approach facilitates access to language technology for billions of English dialect speakers who are traditionally underrepresented.

## 1 Introduction

Dialectal diversity stems from racial, cultural, religious, ethnic, regional, socio-economic, and age-related differences. Considering the increasingly widespread integration of LLMs (Dai et al., 2019; Liu et al., 2019; Raffel et al., 2020) in daily tools, these LLMs should be made invariant to dialectal differences. This is not yet the case, in fact, a significant gap in the performance of LLMs is observed when they are applied to English dialects linguistically distant from Standard American English (SAE) (Jurgens et al., 2017; Blodgett et al., 2018; Kiritchenko and Mohammad, 2018; Ziems et al., 2023b). These discrepancies raise racial, ethnic, and socio-economic concerns for groups that

| Methods | Unseen | |
| | Tasks | Dialects |
|---|---|---|
| Jørgensen et al. (2016) | ✗ | ✗ |
| Blodgett et al. (2018) | ✗ | ✗ |
| Multi-VALUE Ziems et al. (2023b) | ✗ | ✗ |
| TADA Held et al. (2023) | ✓ | ✗ |
| HyperLoRA | ✓ | ✓ |

Table 1: Comparison of previous work in dialectal robustness under zero-shot transfer capabilities to new tasks and new dialects.

are under-represented (Gururangan et al., 2022) in the training corpus of these technologies (Hovy and Spruit, 2016; Blodgett and O'Connor, 2017; Halevy et al., 2021a). Understanding and mitigating these discrepancies are particularly important in avoiding harmful and undesired consequences, which can range from denial of care in commercial healthcare systems (Obermeyer et al., 2019) to racial biases in hate speech detection (Davidson et al., 2019; Sap et al., 2019; Rios, 2020; Halevy et al., 2021b; Zhou et al., 2021).

Previously, dialectal robustness methods have primarily focused on filling the lack of dialect data via manual (Blevins et al., 2016; Blodgett et al., 2018) and weak forms of supervision (Jørgensen et al., 2016; Jurgens et al., 2017), or more recently via synthetic data augmentation (Multi-VALUE; Ziems et al., 2022, 2023b). A shared limitation of these methods is their assumption of available dialectal data for all downstream tasks. In practice, this is unrealistic, as it is already challenging to find annotators in all dialects (Ziems et al., 2023b). Recent work has started to reduce the burden on task-specific dialectal data, such as by training task-agnostic adapters via cross-dialectal alignment (Held et al., 2023). While new dialects are emerging and existing dialects are evolving, the need for data in all dialects remains, as well as

adaptation methods that are resource-efficient and task-agnostic.

To this end, we propose HyperLoRA, an efficient adaptation method to new dialects without the need for additional dialect annotations. In removing this dependency on dialect data, we turn to existing expert knowledge on dialects. Bird (2022) claims that we do not need to bridge the gap in data in settings where expert knowledge is readily available. This assumption of having access to expert knowledge is reasonable because the cost of having a single expert identify the dialect of a speaker is much lesser than hiring annotators from all dialects. Previously, the use of typological features has been successful in removing this gap in the multilingual setting (Ansell et al., 2021). Inspired by this, our work investigates whether this expert knowledge and typological features can be leveraged for dialects as well.

A natural solution in leveraging this expert knowledge is via hypernetworks (Ha et al., 2016), which have exhibited remarkable generalization capabilities in computer vision and NLP (Knyazev et al., 2021; Üstün et al., 2022). Using a hypernetwork, we modulate dialect-specific LoRA (Hu et al., 2021) adapters using typological features for adaptation to target dialects. By isolating the complexity of the typological space to the hypernetwork and by generating dialect-specific LoRA adapters, we minimize the cross-dialectal interference (Wang et al., 2020) in the main model. The hypernetwork is trained on parallel corpora to optimize a morphosyntactic alignment objective in the representation space, which allows HyperLoRA to learn to adapt to dialects independently of the downstream application. This alignment objective is novel, principled, and easy to compute. Most importantly, we find that effectively using expert knowledge can account for 250 annotations per dialect. Finally, we design a metric to evaluate the coverage of dialect features, in order to better understand the limitations of using hypernetworks for zero-shot generalization to dialects.

## 2   Related Work

**Dialectal NLP** When applied to other English dialects, existing language models that primarily focus on Standard American English (SAE) often demonstrate significantly lower performance (Sap et al., 2019; Rios, 2020; Halevy et al., 2021b; Zhou et al., 2021). Previous research has revealed that

prompting LLMs can further degrade the performance on these dialects (Ziems et al., 2023a; Liu et al., 2023). These discrepancies can further reinforce existing power imbalances (Hovy and Spruit, 2016; Bommasani et al., 2021) and bring allocational harm to specific racial, ethnic, and socio-economic communities. This is precisely why the development of dialect robust methods are currently of the utmost importance.

**Transfer Learning** Transfer Learning has become the dominant paradigm in specializing models to target languages and tasks. To this effect, many parameter-efficient fine-tuning (PEFT) (Hu et al., 2021; Houlsby et al., 2019; Zaken et al., 2022) modules have been designed for efficiently adapting Large pretrained Language Models to downstream applications (Pfeiffer et al., 2023). MAD-X (Pfeiffer et al., 2020b) shows that separate task and language adapters can be composed to achieve multi-task cross-lingual transfer. Like MAD-X, TADA (Held et al., 2023) trains dialect-specific adapters separately from task adapters, allowing the adaptation of the SAE-trained model to different dialects in a task-agnostic manner. These works, however, are limited by the need to train an adapter for each language/dialect. To address this shortcoming, several works make use of hypernetworks to generate language-specific adapters from language typological vectors (Ansell et al., 2021) and language identifiers (Üstün et al., 2022), effectively removing the need to train over hundreds of language adapters. In addition to adapters, hypernetworks have also been applied to prompt-tuning (He et al., 2022) and LoRA (Phang et al., 2022). While prior work mainly generates modules for language adaptation, our work is the first to perform dialect adaptation via hypernetworks.

**Cross-lingual alignment** Cross-lingual alignment has been observed in learned representations of multilingual language models (Pires et al., 2019). Alignment is a particularly desirable property enabling task-adapter modules to be shared across languages. Furthermore, cross-lingual alignment methods (Conneau et al., 2018, 2020) are particularly effective when working with highly similar languages, making them suitable for the cross-dialectal setting. Surprisingly, this cross-dialectal setting remains underexplored. In most settings, token-to-token level supervision for alignment is unavailable. Prior works have addressed this by

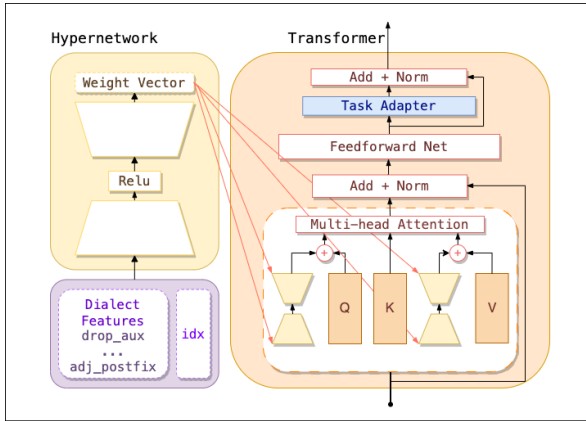

Figure 1: HyperLoRA Architecture. During training, only hypernetwork weights are updated and there is no task adapter in the main model. At inference, the task adapter and its classification head are added.

developing unsupervised methods. In this line of work, a few methods perform cross-lingual alignment by minimizing an approximate Wasserstein distance (Arjovsky et al., 2017; Romanov et al., 2019). Alternatively, prior work has shown that directly optimizing for a relaxed Wasserstein distance using Sinkhorn's Divergence can also be effective for cross-lingual alignment when sufficiently reliable representations are available (Zhang et al., 2017). In our setting, Multi-VALUE provides us with an abundance of pseudo-dialectal training data, which makes it possible for us to design a morphosyntactic alignment objective.

## 3 HyperLoRA

As a first step towards dialectal robustness, Hyper-LoRA enables resource-efficient adaptation to new dialects in a task-agnostic manner. Our approach relies on 4 key ingredients: (1) we support low-resource dialects with expert linguistic knowledge whose information is modeled by (2) a hypernetwork that learns a shared linguistic feature space across dialects. The hypernetwork is trained to generate (3) lightweight LoRA modules with (4) the objective to align dialect and SAE representations by finding the optimal transport plan. Under this optimal transport plan, we can directly plug the LoRA modules into any downstream task.

### 3.1 Dialectal Typology as Expert Knowledge

*"The man I met's girlfriend is a real beauty"*, is what an East Anglian dialect speaker would say instead of *"The girlfriend of the man I met is a real beauty"*. The East Anglian speaker uses a construc-

tion where the possessive marker is appended at the end of the noun phrase. To linguists, this is known as a linguistic feature or linguistic rule that dialect speakers employ at different rates and in different contexts. Experts have found that this feature is not unique to the East Anglian dialect and can be found in many dialects geographically close to the East Anglian dialect, or even in Indian English and in Hong Kong English, with lower levels of pervasiveness. Experts have long studied the intra- and cross-dialectal variations in the lens of these typological features. We follow the intuition of Nerbonne (2009), *defining dialects by their unique sets of correlated dialect features*. These typological feature vectors are readily available on the Electronic Atlas of Varieties of English (eWAVE; Kortmann et al., 2020)[1]. Multi-VALUE applies feature transformations probabilisticially according to their attestation in eWAVE at the following rates: 100% for obligatory features, 60% for features neither pervasive nor rare, 30% for rare features and 0% for no information or attested absence. We follow this procedure. More specifically, we model the space of linguistic features jointly with their aggregation patterns using a neural network and investigate its usefulness for cross-dialectal generalization.

### 3.2 HyperNetworks

We leverage hypernetworks for Low-Rank Adaptation (LoRA). LoRA (Hu et al., 2021) is a fine-tuning approach that keeps the full model parameters fixed and instead updates a low-rank decomposition of the attention matrices. Instead of updating LoRA weights directly, our approach learns the weights of a hypernetwork (Ha et al., 2016), which is then used to generate the appropriate LoRA weights. To our knowledge, we are the first to generate LoRA adapters with a hypernetwork for domain adaptation. We give a detailed outline in Figure 1 for this novel hypernetwork architecture for generating LoRA parameters. Concretely, we lay out the notation for our hypernetwork architecture as follows. Let $D_q^k, U_q^k$ denote the layer $k$ low-rank projections associated with the query, and $D_v^k, U_v^k$, those associated with the value. We use hypernetworks $g$ taking as input $\mathsf{concat}(d, i_{\{q,v\}}^k)$ where $d \in [0,1]^{\#\text{ features}}$ is the dialect feature vector and $i_{\{q,v\}}^k \in \{0, \ldots, 2 \times \#\text{ blocks}\}$ the posi-

---

[1]These vectors can be found at https://github.com/SALT-NLP/multi-value

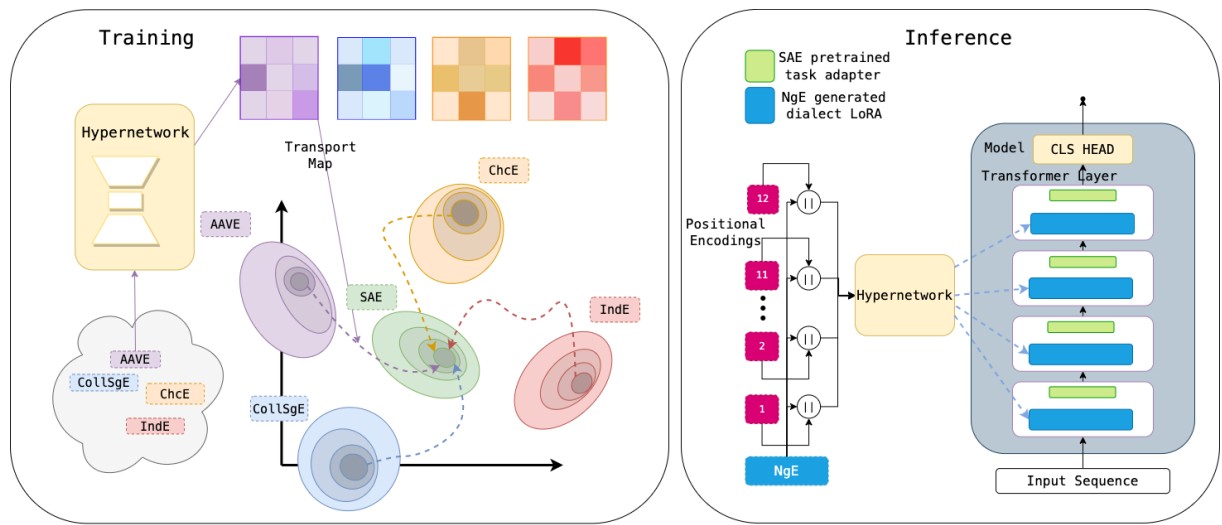

Figure 2: **Training and Inference pipelines**: During training, the hypernetwork learns a mapping from the dialect feature vector to the LoRA adapter weights that perform alignment. During inference, the same hypernetwork is used to generate dialect-specific LoRA adapters from dialect features. At both the training and inference time, we concatenate a positional encoding to the dialect feature to differentiate between transformer blocks, and between the query and the value LoRA adapters.

tional embedding that differentiates between layers and between queries and keys. We use separate hypernetworks for $D_{\{q,v\}}^k$ and $U_{\{q,v\}}^k$. Each hypernetwork is parameterized by weights $W_d, W_u$ denoting the down and up projections respectively. Finally, for $D_{\{q,v\}}$ (similarly for $U_{\{q,v\}}$) the hypernetwork equations can be written as:

$$x = \text{concat}(d, i_{\{q,v\}}^k) \quad (1)$$

$$D_{\{q,v\}}^k, U_{\{q,v\}}^k = g(x), g'(x) \quad (2)$$

and more specifically:

$$D_{\{q,v\}} = \text{MM}(\text{ReLU}(\text{MM}(x, W_d)), W_u) \quad (3)$$

where MM stands for matrix multiplication. Equation 3 also applies to $U_{\{q,v\}}^k$ with its respective weights via a similar calculation. Training HyperLoRA is shown in Figure 2 and Algorithm 1.

### 3.3 Dialect-Specific Low-Rank Adaptation

Previous cross-lingual adaptation methods have focused on a variety of different bottleneck adapter configurations applied after the multi-head attention in the transformer layer (Lialin et al., 2023; Pfeiffer et al., 2020b, 2023). Building upon these efforts, we hypothesize that *adaptation at the attention level can be effective for the morphosyntactic variations present in dialects*. This hypothesis stems from the observation that the self-attention mechanism, known for its sensitivity to syntactic nuances, can better serve syntactical variations

across and within dialects. However, a comprehensive examination of PEFT modules for dialects is needed, which we leave for future work.

### 3.4 Morphosyntactic Alignment

While there is an abundance of sentence parallel bitexts originating from machine translation used for cross-lingual alignment, the equivalent does not exist for English dialects. As a remedy, we employ the rule-based translation system of Multi-VALUE (Ziems et al., 2023b) to generate parallel corpora for all source dialects. While Multi-VALUE evaluation was shown to be predictive of real-world performance (Ziems et al., 2023b), the synthetic nature of this evaluation is a limiation of our work discussed further in the Limitations section.

The Multi-VALUE transformed corpora are only aligned at the sentence level. However, the differences we tackle lie at the morphosyntactic level, which calls for a token-level alignment. To this end, we leverage unsupervised alignment methods discussed in previous work (Zhang et al., 2017; Alvarez-Melis and Jaakkola, 2018). We measure token-level variations via the earth mover's distance, denoted as $\text{W}(\mathbb{P}_{\textbf{DIAL}}, \mathbb{P}_{\textbf{SAE}})$, where $\mathbb{P}_{\textbf{DIAL}}$ represents the distribution of dialect last layer representations, while $\mathbb{P}_{\textbf{SAE}}$ corresponds to the distribution for SAE. The earth mover's distance, or Wasserstein's distance (W), can be approxi-

mated via Sinkhorn's divergence (Feydy et al., 2018) which interpolates between the Wasserstein Distance, and the Maximum Mean Discrepancy (MMD) via the equation:

$$S_\varepsilon(\alpha, \beta) \stackrel{\text{def}}{=} W_\varepsilon(\alpha, \beta) - \frac{1}{2}W_\varepsilon(\alpha, \alpha) - \frac{1}{2}W_\varepsilon(\beta, \beta)$$

Here $W_\varepsilon$ is the computationally-efficient entropy regularized Wasserstein distance (Cuturi, 2013), which is defined as follows:

$$W_\varepsilon(\alpha, \beta) \stackrel{\text{def}}{=} \min_{\pi \in \Pi(\alpha, \beta)} \int_{\mathcal{X} \times \mathcal{Y}} c(x, y) d\pi(x, y) + \varepsilon \text{KL}(\pi, \alpha \otimes \beta)$$

where $x$ and $y$ are the last layer dialect and SAE representations respectively. And similarly $\mathcal{X}$ and $\mathcal{Y}$ are the feature spaces for last layer dialect and SAE representations, respectively. $\pi$ is the coupling that minimizes the cost $c$ of moving mass from distributions $\alpha$ to $\beta$. To compute the Sinkhorn divergence, we use the solver provided by Feydy et al. (2018) with $\varepsilon = 0.05$ and $c$ as the squared error.

---

**Algorithm 1** HyperLoRA Training

**Input:** features $\{d_s\}_{s \in \mathcal{S}}$, sentences $\{x_s\}_{s \in \mathcal{S}}$, SAE representations $h_{\textbf{SAE}}$
Initialize $M$ # Main model
Initialize $g$ # Hypernetwork
**for** training step **do**
    $s \sim S$ # Sample dialect
    $B_s \sim \{x_s\}$ # Sample batch
    $\theta_s \leftarrow g(d_s)$ # LoRA adapter
    **for** $x_s \in B_s$ **do**
        $h_s \leftarrow M(x_s; \theta_s)$ # last hidden states
    **end for**
    loss $\leftarrow S_\varepsilon(\{h_s\}, h_{\textbf{SAE}})$
    backpropagate loss in $g$
**end for**
**Return:** $g$

---

## 4 Experimental Setup

**Datasets** We evaluate our method on 5 dialect transformed variants of GLUE using Multi-VALUE (Ziems et al., 2023b). We choose African American Vernacular English (AAVE), Indian English (IndE), Nigerian English (NgE), Colloquial Singaporean English (CollSgE), and Chicano English (ChcE) as our dialects of focus. AAVE has

been the primary focus of previous works in dialectal robustness. IndE and NgE are widely used dialects by over a hundred million of speakers. CollSgE has shown to be a particularly difficult dialectal shift (Ziems et al., 2023b) sharing little linguistic features with mainstream SAE, and with many unique features in CollSgE alone. ChcE on the other hand is particularly close to SAE. In our experiments, we focus on these 5 dialects. Later, in our ablation studies, we will explore training HyperLoRA on other dialects closer to CollSgE to study the impact of dialects used at training time.

**Training Details** For all experiments, we use a pretrained RoBERTa Base (Liu et al., 2019) as the backbone model. For the training of Hyper-LoRA, we use 1000 WiC (Pilehvar and Camacho-Collados, 2019) examples from each source dialect. At inference time, we plug the generated LoRA module from the learned hypernetwork in the backbone model with appropriate task-specific adapters and their associated classification heads. We train HyperLoRA with 4 source dialects using the Adam (Kingma and Ba, 2017) optimizer with a learning rate of 3e-5, with a linear scheduler, and using a batch size of 16 for 50 epochs. We load the model with the lowest loss at the end of training. These hyperparameters have been selected via a grid search over learning rates of 1e-5, 3e-5, and 1e-4, batch sizes of 16, 32, and 64, and between 30 and 50 epochs. For task-specific adapters, we directly utilize readily available GLUE adapters from Adapterhub (Pfeiffer et al., 2020a). In all of our experiments, HyperLoRA is trained and evaluated in a zero-shot fashion. For each unseen dialect (e.g., A), we train HyperLoRA using the remaining dialects (B, C, D, E) and evaluate its dialectal robustness against A. For example, in Figure 2, HyperLoRA is trained on AAVE, NgE, ChcE, and IndE, and evaluated on the target dialect CollSgE.

**Baselines** In benchmarking HyperLoRA, we evaluate its (1) resource efficiency against current task-agnostic dialect methods, its (2) dialectal robustness against models trained for SAE, and its (3) ability to effectively utilize expert knowledge for adapting to new dialects. For each of these research questions, we establish a suitable baseline. To address the resource efficiency of our method, we compare HyperLoRA with **TADA** (Held et al., 2023) trained on varying numbers of examples from the target dialect. More specifically, for each

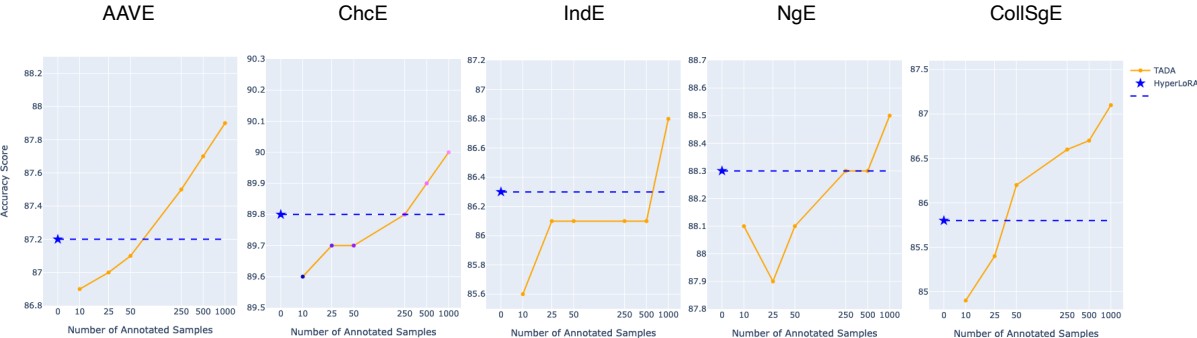

Figure 3: **QQP Performance under Few-shot Evaluation**: The x-axis denotes the number of examples of the target dialect being used in the model on a log scale. We use a blue star (and the scattered line) to denote the performance of HyperLoRA, while the orange line curve shows the performance of TADA using increasingly more annotated samples. The cost of training scales linearly with the number of annotated samples.

$k \in [10, 25, 50, 250, 500, 1000]$, we train TADA on $k$ WiC samples. We follow TADA to use 1000 examples and keep the remaining training details unchanged. To highlight the dialectal robustness of HyperLoRA, we implement a simple adapter baseline, which we denote by **SAE**. Using RoBERTa-Base as our backbone, we add task-specific adapters trained on the original GLUE dataset. Similarly to HyperLoRA, this is a zero-shot baseline. Finally, we establish a baseline that does not utilize expert knowledge. To do this, we remove the hypernetwork component of HyperLoRA, keeping **LoRA** modules and our alignment loss. As opposed to HyperLoRA, these LoRA modules are cross-dialectal. We train and evaluate both Hyper-LoRA and LoRA in the same zero-shot manner.

## 5 Experimental Results

### 5.1 Efficient Adaptation to Unseen Dialects

First, we highlight the efficiency of using expert knowledge in adapting to new dialects. For the sake of simplicity, we restrict our evaluation to Quora Question Pairs (QQP), which is one of the tasks with the least variability in performance.

In Figure 3, we show QQP performances across all 5 dialects. HyperLoRA finds competitive performance at a much lower cost, showing comparable performance to TADA trained on $\approx 250$ dialect samples. For AAVE, and CollSgE, this is lower, around 50 and 25 respectively. This observation highlights the value of expert linguistic knowledge for dialect adaptation, as it can be equivalent to having 250 annotated samples per dialect—a substantial benefit. The significance of this finding becomes evident when considering the vast number of existing dialects, which exceeds 70, and

the potential emergence of new ones. Acquiring 250 annotated samples for each dialect can be prohibitively expensive and challenging in terms of finding suitable annotators (Ziems et al., 2023b). We acknowledge that while HyperLoRA may not completely bridge the performance gap, it effectively addresses the trade-off between performance and resource constraints without any dialect examples. Consequently, it provides a valuable degree of robustness at an almost negligible cost.

### 5.2 Zero-Shot Transfer Results

To evaluate the dialectal robustness of HyperLoRA, we compare HyperLoRA to the SAE baseline across all 5 dialects in Table 2. We observe that our method generally achieves higher performance over the SAE baseline, with the exception of RTE. Noticeably, HyperLoRA achieves higher performance on more than 4 out of 7 tasks. In analyzing these results, we find that COLA, RTE, and SST2 suffer from large variability in performance. On the remaining tasks, that is MNLI, QNLI, QQP, and STSB, HyperLoRA achieves the best or competitive performance. As a whole, there is an improvement of 1.7% in mean performance for AAVE and 0.8% in mean performance for NgE.

In the case of ChcE, our approach fails to bring a mean performance improvement. It is worth noting that the authors of Multi-VALUE (Ziems et al., 2023b) also encountered a similar outcome when training on ChcE instead of SAE. This lack of improvement can be attributed to the striking similarities between ChcE and Colloquial American English. This set of experiments takes into account the potential variability in the differences between the source dialects used to train HyperLoRA and

| Unseen Dialect | COLA Orig. | COLA Ours | MNLI Orig. | MNLI Ours | QNLI Orig. | QNLI Ours | RTE Orig. | RTE Ours | QQP Orig. | QQP Ours | SST2 Orig. | SST2 Ours | STSB Orig. | STSB Ours | Mean Orig. | Mean Ours |
|---|---|---|---|---|---|---|---|---|---|---|---|---|---|---|---|---|
| AAVE | -0.02 | **10.5**$_+$ | 83.7 | **83.8** | **90.5** | **90.5** | **68.9** | 68.4 | 87.0 | **87.2** | 92.8 | **93.5** | 88.5 | **88.7** | 73.0 | **74.7** |
| ChcE | 30.7 | **31.0** | **86.3** | **86.3** | 93.0 | **93.1** | **68.5** | 66.8 | 89.6 | **89.8** | **93.5** | 93.1 | **90.1** | **90.1** | **78.8** | 78.6 |
| IndE | **19.4** | 18.9 | 82.6 | **82.9** | **89.4** | 89.3 | 64.2 | **65.0** | 86.1 | **86.3** | 92.0 | **92.2** | 88.1 | **88.7**$_+$ | 74.5 | **74.8** |
| NgE | 24.7 | **26.6** | 84.6 | **84.7** | 90.8 | **91.0** | 64.2 | **66.1** | 88.2 | **88.3** | 92.0 | **92.6** | 89.5 | 89.5 | 76.2 | **77.0** |
| CollSgE | 4.5 | **8.0** | 82.0 | **82.2** | **88.3** | 88.2 | **66.4** | 64.6 | 85.0 | **85.8**$_+$ | **91.6** | 91.1 | 87.5 | **87.7** | 72.1 | **72.5** |

Table 2: **Zero-shot performance on GLUE.** For each task, we report the SAE Task Adapter performance (Orig.) and the HyperLoRA performance (Ours). We report Matthew's Correlation score for COLA, the Pearson-Spearman correlation score for STS-B, and accuracy for the rest. Via a paired bootstrap test at $\alpha = 0.05$, we label significant improvements for each task with $+$. There was no significant drop in performance.

| Methods | CollSgE Glue Performance | | | | | | | |
|---|---|---|---|---|---|---|---|---|
| Model | COLA | MNLI | QNLI | RTE | QQP | SST2 | STS-B | Mean |
| SAE | 4.5 | 82.0 | **88.3** | **66.4** | 85 | **91.6** | 87.5 | 72.1 |
| LoRA | 0.7 | 82.0 | **88.3** | 64.6 | 85 | 91.0 | 87.5 | 71.3 |
| HyperLoRA | **8.0**$_†$ (+7.3) | **82.2** | 88.2 | 64.6 | **85.8**$_{++}$ (+0.8) | 91.1 | **87.7** | **72.5** |

Table 3: **CollSgE GLUE Performance** With RoBERTa Base as our base model, we compare adding SAE-trained task adapters, adding SAE task adapters and LoRA, and adding SAE task adapters and HyperLoRA. Both LoRA and HyperLoRA are trained on AAVE, Chicano English, Nigerian English, and Indian English. For each task, we run a paired bootstrap test with $\alpha = 0.05$ and label significant improvements w.r.t. the SAE Task Adapter with $+$ and w.r.t. the LoRA baseline with $†$. There was no significant drop in performance.

the dialects HyperLoRA is evaluated on. This variability can explain the differences in performance gain across dialects.

As a plug-and-play module that can be readily used by any community, HyperLoRA has the potential to improve the robustness of the SAE-trained backbone model regardless of dialect.

### 5.3 Effectiveness of Expert Knowledge

In order to validate the contribution of expert knowledge, we compare HyperLoRA with the LoRA baseline. We report the results in Table 3. We observe that training cross-dialectal LoRA adapters can negatively impact GLUE performance. When compared to the naive SAE baseline, LoRA demonstrates poorer performance with a decrease of 3.8% and 1.8% on COLA and RTE, respectively. For HyperLoRA, we have found that although there is still a slight decrease in RTE performance -1.8%, it proves to be superior over the SAE baseline. Specifically, HyperLoRA brings improvements of 3.5% on COLA and 0.8% on QQP. Through a paired bootstrap test, we verify that the drop in RTE performance is not statistically significant, while the improvements on COLA and QQP are statistically significant. In conclusion, our findings suggest that employing a hypernetwork to minimize negative interference, along with lever-

aging expert knowledge, proves to be an effective strategy for improving cross-dialectal transfer.

## 6 Ablation Analyses

### 6.1 Morphosyntactic Alignment

To understand the effectiveness of our morphosyntactic objective, we return to TADA's setup and modify its alignment objective to our Sinkhorn Divergence. For both TADA and our alignment objective, we train dialect-specific adapters for AAVE using 1000 parallel samples from the SAE WiC dataset and the Multi-VALUE transformed AAVE Wic Dataset. We evaluate these adapters on the GLUE benchmark and report the results in Table 4.

We observe that both TADA and our alignment objective outperform the naive SAE task adapter. While our strategy achieves +0.7% on COLA and -1.8% on RTE comparatively to TADA, we verify through a paired bootstrap test and find that these differences are not statistically significant. Therefore, with no significant difference in performance, our Sinkhorn divergence-based morphosyntactic alignment objective presents a well-founded optimization problem that can be efficiently solved. It offers desirable convergence guarantees, eliminating the necessity for additional heuristics employed in the adversarial training approach in TADA.

| Methods | Adapters | | AAVE Glue Performance | | | | | | | |
| Model | Dialect | Task | COLA | MNLI | QNLI | RTE | QQP | SST2 | STS-B | Mean |
|---|---|---|---|---|---|---|---|---|---|---|
| SAE | ✗ | ✓ | -0.02 | 83.7 | 90.5 | 68.9 | 87.0 | 92.8 | 88.5 | 73.0 |
| TADA | ✓ | ✓ | 24.5 | 84.8 | 91.7 | 70.4 | 88.1 | 93.0 | 89.6 | 77.4 |
| Sinkhorn | ✓ | ✓ | 25.2 | 84.7 | 91.5 | 68.6 | 88.1 | 93.3 | 89.4 | 77.3 |

Table 4: **Alignment Objectives**: We compare the cross-dialectal alignment objective TADA with our objective based on the Sinkhorn divergence. For both objectives, we train dialect adapters on AAVE data, and evaluate it on AAVE GLUE tasks. We run a paired bootstrap test at $\alpha = 0.05$ but find no significant difference between TADA and Sinkhorn performances.

| | | | CollSgE Glue Performance | | | | | | | |
| Source Dialect | L1 dist | Coverage | COLA | MNLI | QNLI | RTE | QQP | SST2 | STS-B | Mean |
|---|---|---|---|---|---|---|---|---|---|---|
| SAE | | | 4.5 | 82.0 | **88.3** | 66.4 | 85.0 | **91.6** | 87.5 | 72.1 |
| MalaE, MaltE, JamE, IndSAE | 0.219 | 87.8 | 7.6 | 82.1 | 88.2 | **67.2** | 85.7₊ | 90.7 | **88.0** | **72.8** |
| CapeE, FijiAE, MaltE, SriLE | 0.209 | 65.7 | 7.4 | **82.2** | **88.3** | 65.7 | 85.7₊ | 90.7 | 87.9 | 72.6 |
| NgE, AAVE, IndE, ChcE | 0.257 | 81.3 | **8.0** | **82.2** | 88.2 | 64.6 | **85.8₊** | 91.1 | 87.7 | 72.5 |

Table 5: **Impact of Source Dialects**: We compare CollSgE performance when training HyperLoRA on different source dialects. Typically low average L1 distance and higher coverage indicate that the source dialects are closer to the target dialect. We label significant improvements in performance over SAE with +.

## 6.2 Impact of Source Dialects

We study the impact of the source dialects more closely by analyzing the distinctiveness of the new dialect at test time with respect to the source dialects used for training. This distinctiveness of dialect feature sets is natural, in fact, it is commonly known in dialectology that some features contradict each other (Nerbonne, 2009). Commonly used metrics to measure dialect differences are the geographical distance and the Manhattan distance applied dialect feature vectors (Ziems et al., 2023b). However, these metrics are not directly suited for the multi-source setting. To this effect, we develop a metric for feature coverage, as follows. We hypothesize that HyperLoRA performs best on new dialects when most of the linguistic features of the new dialect have been seen during training.

$$\text{Coverage} = 1 - \frac{\|[(\sum_{s \in \mathcal{S}} d_s) - d_t]_-\|_1}{\|d_t\|_1} \quad (4)$$

where $d_s$ and $d_t$ are the linguistic feature vectors for a source dialect $s$ and the target dialect $t$, respectively. $\mathcal{S}$ represents the set of source dialects. Our metric effectively computes the percentage of weighted features in the target dialect that are covered by dialects in $\mathcal{S}$.

To measure the impact of source dialects, we compute the average Manhattan distance and the coverage score for all combinations of 4 dialects that are different from the target dialect. For

CollSgE, we find that the set (CapeE, FijiAE, MaltE, SriLE) attains the lowest Manhattan distance, but also a low coverage score. Moving up in the pareto frontier, the set (MalaE, MaltE, JamE, IndSAE) attains a low Manhattan distance, but high coverage score. We train HyperLoRA for these two sets of source dialects and compare the performance to our previous experiment (Section 5). We report the results in Table 5.

We find that both lower average Manhattan distance and larger feature coverage can contribute to performance improvement on the target dialect. Specifically, simultaneously decreasing the Manhattan distances and improving the feature coverage can lead to an improvement of +2.6% on RTE (from NgE, AAVE, IndE, ChcE to MalaE, MaltE, JamE, IndSAE). Overall, when the new dialect is particularly close in Manhattan distance and largely covered by the source dialects, we observe HyperLoRA can lead to the highest performance, with an improvement of +0.7% on mean performance, compared to the SAE baseline.

Based on these findings, we demonstrate that when computational resources are limited, employing these heuristics offers a straightforward and efficient strategy for selecting the source dialects when addressing evolving dialects.

## 7 Conclusion

In this paper, we propose HyperLoRA, a task-agnostic, light-weighted, and highly scalable di-

alect adaptation method. Where only accessing expert knowledge about dialects, we show that Hyper-LoRA can lead to robustness improvement against unseen dialects on the GLUE benchmark, across five dialects. At inference time, HyperLoRA does not require any dialect data, which makes it widely applicable in resource and compute-constrained settings. Furthermore, HyperLoRA is trained with a data volume that can be easily replaced by manually translated dialect corpora. This resource and computational efficiency greatly facilitate the appropriation of language technologies within small but diverse communities[2]. Finally, by generating LoRA adapters using a lightweight hypernetwork, our approach is highly portable to LLMs with less than 0.5% additional parameters and without any additional inference latency. These aspects enable HyperLoRA to achieve a favorable tradeoff between the training and inference cost and dialectal robustness. To sum up, HyperLoRA holds great potential to enable billions of traditionally underrepresented English dialect speakers to access language technology using their preferred languages.

## Limitations

HyperLoRA is trained on pseudo-dialects obtained using the Multi-VALUE (Ziems et al., 2023b) transformation rules, which are synthetic dialectal shifts that focus on morphology and syntax-related differences. It is important to note that these shifts do not encompass the entirety of possible variations found in real-world dialects. Therefore, we encourage future research to address this limitation and explore other naturally occuring variations associated with dialects such as lexical differences, topical shifts and register shifts. Additionally, while HyperLoRA can utilize any linguistic vector that provides a more detailed characterization of dialects during the testing phase, we did not conduct a sensitivity analysis for these vectors. This lack of guarantee can pose challenges since real-world dialectal variations are often much more nuanced and intricate.

Furthermore, our work does not include a comprehensive comparison of various parameter-efficient fine-tuning techniques for dialect adaptation. We encourage further research to delve into this area and explore it.

Finally, all of our experiments primarily focus on encoder-only LLMs. As a result, this creates an ex-

periment gap where we are unable to verify the performance of our method on encoder-decoder, and decoder-only architectures. Future work should fill the gap and further explore task-agnostic dialect adaptation solutions for models with these alternate architectures.

## Ethics Statement

As highlighted in our limitations, we acknowledge that we are unable to offer guarantees regarding the usage of HyperLoRA in communities where intra-dialectal variations are prevalent. This limitation stems from the fact that dialects are not uniform entities and encompass diverse variations. Therefore, it is crucial for members of these dialect communities to take necessary precautions when applying HyperLoRA to their use cases.

## Acknowledgement

We would like to thank the anonymous reviewers and SALT lab members for their valuable feedback. This work was partially sponsored by the Defense Advanced Research Project Agency (DARPA) grant HR00112290103/HR0011260656, and NSF grant IIS-2247357 and IIS-2308994.

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

## A  Alignment losses

As an attempt to explain the closeness in performance in our alignment objective and TADA's alignment objective, we take a closer look at TADA's morphosyntactic alignment objective. TADA solves the alignment problem via adversarial training, where the critic optimizes:

$$\max_{\text{Adv}} \mathbb{E}\left[\ell_{\text{adv}}\right] = \max_{\theta} \mathbb{E}_{d\sim\mathbb{P}_{\textbf{DIAL}}}\left[\text{Adv}(d;\theta)\right]$$
$$- \mathbb{E}_{s\sim\mathbb{P}_{\textbf{SAE}}}\left[\text{Adv}(s;\theta)\right]$$

assume Adv is K-Lipschitz,

$$\approx \frac{1}{K}\sup_{\|\text{Adv}\|_{L}\leq K}\mathbb{E}_{d\sim\mathbb{P}_{\textbf{DIAL}}}\left[\text{Adv}(d;\theta)\right]$$
$$- \mathbb{E}_{s\sim\mathbb{P}_{\textbf{SAE}}}\left[\text{Adv}(s;\theta)\right]$$

When $c = \ell_2$,

$$= \text{W}(\mathbb{P}_{\textbf{DIAL}}, \mathbb{P}_{\textbf{SAE}})$$

Under the $\ell_2$ ground distance, the last step follows from the Kantorovich-Rubinstein duality (Villani, 2008). Briefly, when the objective of the critic reaches optimality, it approximates the Wasserstein distance up to scaling factor $K$, while the generator minimizes this approximate distance. As such, we have shown that TADA aims to minimize the same mathematical objective.

Our alignment objective is independent of the chosen ground distance, as opposed to the dual problem used by WGAN that only holds when the ground distance is the $\ell_2$ distance. Using Sinkhorn's divergence, we do not need to introduce an adversarial training procedure that relies on the optimization and the approximation power of a critic network. We understand that this is not a direct comparison as TADA also includes a contrastive sequence loss, thus putting a higher weight on the **CLS** token.

## B  Dialectal Differences

To quantify how much of our test sets are being modified by applying Multi-VALUE, we compute the percentage of entries that have been transformed for each test set in figure 8. On average, for each dialect, we have over 88% transformed entries except for Chicano English. This is expected, as Chicano English shares many similarities with Colloquial American English. In the case of Colloquial Singaporean English, the entries are almost always transformed by Multi-VALUE. It is difficult in practice to get a precise estimate of these differences as dialect variations do not fit in deterministic baskets, instead different features are utilized at different rates.

| Models | | | CollSgE STS-B |
| Source Dialects | L1 dist | Coverage | Performance |
| CapeE, FijiAE, FijiBE, MalaE | 0.228 | 0.866 | 87.97 |
| JamE, aave, AppE, FijiBE | 0.287 | 0.815 | 87.84 |
| JamE, CapeE, MaltE, AbEng | 0.231 | 0.837 | 88.03 |
| JamE, FijiAE, IndSAE, AbEng | 0.238 | 0.844 | 87.99 |
| SriLE, aave, MalaE, AbEng | 0.257 | 0.871 | 87.98 |
| SriLE, IndE, AppE, FijiAE | 0.246 | 0.675 | 87.87 |
| SriLE, IndE, AppE, IndSAE | 0.253 | 0.744 | 87.89 |
| SriLE, NgE, AppE, FijiBE | 0.268 | 0.777 | 87.89 |
| MalaE, MaltE, JamE, IndSAE | 0.219 | 0.878 | 88.03 |
| CapeE, FijiAE, MaltE, SriLE | 0.209 | 0.657 | 87.88 |

Table 6: **CollSgE STS-B Performance** with Hyper-LoRA trained on different source dialects. We report both the L1 distance and the coverage metric.

Furthermore, the applied features to the test sets are diverse. In table 9, we find that across all dialects, a large majority of rules are being applied to the test sets.

## C  Different Source Dialects

Our ablation study focuses on few source dialect combinations. As a result, drawing correlations risk being misleading given the relatively small sample of experiments we have at the moment. We report additional experiments for our ablation study on the impact of source dialects in table 6 and figure 4. In these additional experiments for CollSgE STS-B, training on source dialects with high coverage score and low L1 distance maintains overall best performance.

## D  Computational and Parameter Efficiency

Parameter costs for HyperLoRA are reported in Table 7. $d$ and $t$ mark the dependence on the number of dialects and the number of tasks, respectively. We compare HyperLoRA to Multi-VALUE (Ziems et al., 2023b) and TADA (Held et al., 2023). The Multi-VALUE models are standard fine-tuning and adapter tuning methods applied directly to the dialect transformed task data.

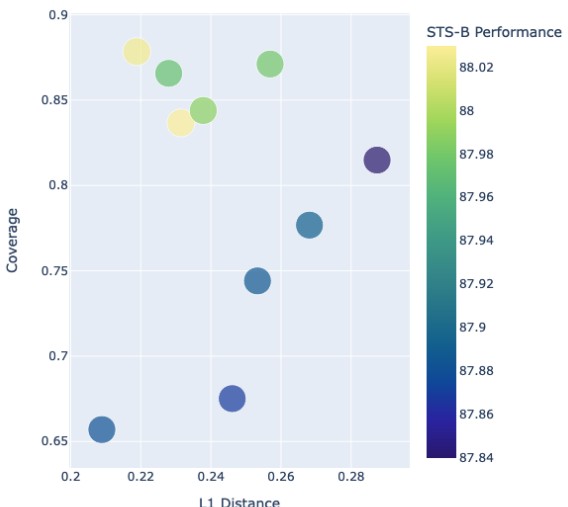

Figure 4: Performance of HyperLoRA trained on different source dialects with respect to L1 distance and Coverage metric

| Model | Approach | # Params |
| --- | --- | --- |
| MultiVALUE | Fine-tuning | $d \times t \times 125M$ |
| | Adapter | $d \times t \times 1.2M$ |
| TADA | Adapter | $d \times 1.5M$ |
| HyperLoRA | HyperLoRA | 225K |

Table 7: Computational efficiency with respect to the number of trainable parameters for MultiVALUE, TADA, and HyperLoRA. All these reported values use a RoBERTa Base model as the base model. TADA includes a critic network.

HyperLoRA is extremely lightweight. We have experimented with more complex architectures which did not show further improvements in performance. We hypothesize this is due to the fact that the space of linguistic features is both simple and has low intrinsic dimension.

## E  LoRA vs Adapters

In a previous iteration of the paper, we investigated the use of Hyperformer++ (Mahabadi et al., 2021) as our hypernetwork instead of HyperLoRA. We present our results in table 10. What we find is that bottleneck adapters are typically worse than LoRA adapters in the zero-shot setting.

| Dialect | Percentage of Transformed Entries | | | | | | | |
|---------|------|------|------|------|------|------|-------|------|
| | COLA | MNLI | QNLI | RTE | QQP | SST2 | STS-B | Mean |
| AAVE | 95.2 | 93.6 | 95.0 | 99.3 | 97.0 | 93.2 | 91.8 | 95.0 |
| ChcE | 55.3 | 59.0 | 26.4 | 74.0 | 41.4 | 57.9 | 37.1 | 50.1 |
| IndE | 98.8 | 96.8 | 99.6 | 100 | 98.8 | 97.1 | 99.6 | 98.7 |
| NgE | 82.6 | 87.5 | 84.5 | 98.6 | 82.2 | 91.3 | 91.2 | 88.3 |
| CollSgE | 99.7 | 97.6 | 99.5 | 100 | 99.7 | 97.1 | 99.8 | 99.1 |

Table 8: **Dialectal Differences** Percentage of transformed entries for each test set.

| Dialect | Total Features | Number of Applied Features | | | | | | |
|---------|----------------|------|------|------|-----|-----|------|-------|
| | | COLA | MNLI | QNLI | RTE | QQP | SST2 | STS-B |
| AAVE | 118 | 92 | 109 | 91 | 86 | 110 | 89 | 92 |
| ChcE | 30 | 23 | 28 | 24 | 25 | 28 | 22 | 24 |
| IndE | 90 | 71 | 85 | 77 | 68 | 84 | 74 | 73 |
| NgE | 45 | 34 | 42 | 32 | 35 | 42 | 34 | 32 |
| CollSgE | 67 | 58 | 63 | 54 | 51 | 63 | 54 | 55 |

Table 9: **Dialectal Differences** Number of applied transformations for each test set.

| Methods | | NgE Glue Performance | | | | | | | |
|---------|------------------|------|------|------|------|------|------|-------|------|
| Model | Trainable Params. | COLA | MNLI | QNLI | RTE | QQP | SST2 | STS-B | Mean |
| SAE Task Adapter | 0 | 24.7 | 84.6 | 90.8 | 64.2 | 88.2 | 92.0 | 89.5 | 76.2 |
| Adapter | 1.1M | 23.8 | 83.8 | 89.9 | 66.7 | 86.9 | 91.7 | 89.0 | 75.9 |
| LoRA | $295K$ | 25.6 | 84.6 | 90.8 | 65.3 | 88.2 | 92.4 | 89.4 | 76.6 |
| Hyperformer++ | 1M | 20.3 | 83.2 | 87.6 | 63.9 | 88.3 | 91.9 | 88.7 | 74.8 |
| HyperLoRA | $225K$ | 26.6 | 84.7 | 91.0 | 66.1 | 88.3 | 92.5 | 89.5 | 77.0 |

Table 10: **Zero-shot NgE GLUE Performance** RoBERTa-Base model adapters and LoRA. We compare adapter models to LoRA models, and in particular, Hyperformer++ to HyperLoRA. LoRA, Adapter, Hyperformer++, and HyperLoRA are trained using our alignment objective.