# OpenReview forum: "Task-Agnostic Low-Rank Adapters for Unseen English Dialects"
_EMNLP/2023/Conference — EMNLP 2023 Main_

### Official Review · Reviewer_wWbU · 2023-08-04

**Soundness:** 4

**Excitement:**

4: Strong: This paper deepens the understanding of some phenomenon or lowers the barriers to an existing research direction.

**Paper Topic And Main Contributions:**

The paper addresses the problem of dialectal diversity in large language models (LLMs) and proposes a task-agnostic adaptation method called HyperLoRA. The main contribution of the paper is the use of expert linguistic knowledge and hyper-networks to enable resource-efficient adaptation (by LoRA) to unseen dialects. The paper shows that HyperLoRA achieves competitive performance across multiple dialects in a zero-shot setting.

**Questions For The Authors:**

1) Can you provide more insights into the limitations and potential challenges of the proposed method, especially regarding the use of pseudo-dialects and the coverage of linguistic features, what kind of risk might have with the pseudo-dialects?

2) Have you explored the effectiveness of HyperLoRA on models with different architectures? It would be helpful to improve the depth of the ablation studies to evaluate the contribution of each component of the model.


**Reasons To Accept:**

1) The paper addresses an important problem of dialectal diversity in language models, as the disproportionality in the representation of Standard American English in LLMs is a well-known issue and it's crucial for ensuring inclusivity and fairness in natural language processing applications.

2) The proposed method, HyperLoRA, leverages expert knowledge and hyper-networks to achieve resource-efficient adaptation to unseen dialects. This approach is innovative and shows promising results and effectively deals with the scarcity of annotated dialect data.

3) The paper provides a solid evaluation of HyperLoRA on the GLUE benchmark, demonstrating its effectiveness in improving dialectal robustness across multiple tasks and dialects, the comparison with relevant baselines strengthens the validity of the proposed method.

4) Comprehensive analysis of the method's task-agnostic nature and its implications for scalability and generalizability to unseen dialects.

5) The paper is well-written and clearly presents the problem, methodology, and experimental results. The Figures are drawn in a nice clear style.



**Reasons To Reject:**

1) The paper lacks a comprehensive comparison with other dialect adaptation methods, which could provide a better understanding of the state-of-the-art in this area.
2) The paper does not explore the effectiveness of HyperLoRA on models with different architectures. Ablation study is not conducted to evaluate the contribution of each component, which could limit the generalizability of the proposed method.


**Reproducibility:**

4: Could mostly reproduce the results, but there may be some variation because of sample variance or minor variations in their interpretation of the protocol or method.

**Reviewer Confidence:**

5: Positive that my evaluation is correct. I read the paper very carefully and I am very familiar with related work.

---

> ### Author Rebuttal · Authors · 2023-08-29
>
> Thank you for your thoughtful feedback and for highlighting in detail the strengths of our paper: “addresses an important problem”, “the approach is innovative”, “promising results”, and “comprehensive analysis”.
>
> **Question 1: Can you provide more insights into the limitations and potential challenges of the proposed method, especially regarding the use of pseudo-dialects and the coverage of linguistic features, what kind of risk might have with the pseudo-dialects?**
>
> Pseudo-dialects obtained via Multi-VALUE [1] are limited to variations in syntactic structure and morphology. This presents important challenges when the dialect of interest contains significant lexical variations. This is the case of pidgins and creoles where it is common to borrow words from another language, or to use slang. As these words are not featured in the vocabulary of these pseudo-dialects.
>
> Because Multi-VALUE is compiled via oral interviews with native speakers, the employed orthographic conventions by linguists may not accurately represent the speaker’s own conventions. This matter becomes even more complex when the dialect is not standardized.
>
> These unaccounted variations create discrepancies in the performance of pseudo-dialects and real dialects. We encourage the development of better tools to account for these differences. Whenever possible, we urge users to properly validate user-facing models with real-world data.
>
> [1] Caleb Ziems, William Held, Jingfeng Yang, Jwala Dhamala, Rahul Gupta, and Diyi Yang. 2023. Multi-VALUE: A Framework for Cross-Dialectal English NLP. ACL 2023.
>
> **Question 2: Have you explored the effectiveness of HyperLoRA on models with different architectures? It would be helpful to improve the depth of the ablation studies to evaluate the contribution of each component of the model.**
>
> The applicability of this method to models other than RoBERTa-Base is important to us and we understand that it currently is a limitation in our work. Experiments for HyperLoRA are currently limited to encoder-only models and do not account for encoder decoder, or decoder-only models. We did not find the time to complete these experiments. We leave the evaluation of HyperLoRA and the development of dialect adaptation methods on these other architectures as future work.

---

### Official Review · Reviewer_ZQT3 · 2023-08-07

**Soundness:** 4

**Excitement:**

4: Strong: This paper deepens the understanding of some phenomenon or lowers the barriers to an existing research direction.

**Paper Topic And Main Contributions:**

This paper deals with the task of adapting language models to new unseen dialects in a resource-efficient way. The method uses hypernetworks to learn a mapping from linguistic dialectal features to LoRA adapter weights during training. By doing this, it alleviates the need for human-annotated dialectal data. The paper evaluates the developed framework on 5 dialectal variations of the GLUE benchmark and uses pre-trained RoBERTa as its backbone. The framework is mostly comparable or slightly better than using task adapters, and is resource-efficient in that it does not need human-annotated data.

**Reasons To Accept:**

1) Experimental Setup: The experimental section is detailed and the hyperparameters are clearly specified. The results are tested for statistical significance and the code will be released.

2) The method alleviates the need for human-annotated samples which is relatively harder to obtain and presents a way to leverage linguistic features instead. In general, the method of leveraging hypernetworks to learn LoRa weights is unique to the best of my knowledge.

3) Ablations and discussion: The ablations section has interesting insights, for example, Section 6.2 where the paper discusses how a larger feature coverage and proximity to a source dialect improves performance. While intuitive, its nice to quantify it and confirm the intuition. It also simplifies the applicability of the method and broadens its scope.

**Reasons To Reject:**

I don't have major reasons of rejection and these are minor points :

1) Related Work & Writing: The related work section might benefit by including information on hypernetworks and LoRA (maybe in the transfer learning section?). Even though I know of these concepts, it took revisiting to understand the framework presented in the paper. In general, its a good idea to make the paper self-contained, and including information strictly relevant to the paper might help with that.

**Reproducibility:**

5: Could easily reproduce the results.

**Reviewer Confidence:**

3: Pretty sure, but there's a chance I missed something. Although I have a good feel for this area in general, I did not carefully check the paper's details, e.g., the math, experimental design, or novelty.

---

> ### Author Rebuttal · Authors · 2023-08-29
>
> Thank you for the acknowledgement of HyperLoRA’s role in “alleviating the need for human-annotated dialectal data”, for recognizing that our ablation study “simplifies the applicability and broadens the scope” of our method, and for your appreciation of our contributions.
>
> We agree with you that the paper should be self-contained. To this effect, we will revise (section 3.2) to include HyperNetworks and Low Rank Adaptation. And we elaborate:
>
> > We leverage hypernetworks for Low-Rank Adaptation .“LoRA [1] is a fine-tuning approach that keeps the full model parameters fixed and instead updates a low-rank decomposition of the attention matrices. Instead of updating LoRA weights directly, our approach learns the weights of a hypernetwork [2], which is then used to generate the appropriate LoRA weights.”
>
> [1] Edward J. Hu, Yelong Shen, Phillip Wallis, Zeyuan Allen-Zhu, Yuanzhi Li, Shean Wang, Lu Wang, and Weizhu Chen. 2021. Lora: Low-rank adaptation of large language models.
>
> [2] David Ha, Andrew Dai, and Quoc V. Le. 2016. Hypernetworks.

---

### Official Review · Reviewer_i59w · 2023-08-11

**Soundness:** 4

**Excitement:**

4: Strong: This paper deepens the understanding of some phenomenon or lowers the barriers to an existing research direction.

**Missing References:**

N/A

**Paper Topic And Main Contributions:**

This paper proposes HyperLoRA, a parameter-efficient technique to adapt LLMs to unseen dialects of English using a feature vector obtained from expert knowledge instead of any annotated data in the target dialect. The core technique is using hypernetworks to generate dialect-specific LoRA modules in a way that can be applied to unseen dialects. Similarly to LoRA, the technique is widely applicable.

The paper provides a number of baselines and ablations that test various hypothetical benefits of their approach.

The authors have promised to publicly release their code.

The paper takes steps to call out and address some limitations, and includes a thoughtful ethics statement.

**Questions For The Authors:**

Question A) Related to my main point in Reasons To Reject, is there additional evidence you can provide that testing on synthetic data in this case still allows us to draw conclusions about a hypothetical real-world scenario?

Question B) The introduction mentions dialects without standardized writing systems as a specific use-case where zero-shot methods are especially beneficial. Are any of the dialects you experimented with examples of this? I imagine that those with non-standardized writing systems may have lower token overlap with SAE, which could make it more difficult to leverage the LLM's pretraining.

Question C) Your technique is clearly applicable in the setting where no labeled data is available, but what about when a small amount is? If your method is able to make use of some labeled examples, it would be interesting to see that compared to Figure 3.

Question D) Did you investigate how much your various test sets were actually modified by applying Multi-VALUE? There are many ways to do so; one would be calculating a BLEU score of the transformed corpus against the SAE corpus. I think this could be useful to get an approximate idea of available headroom: if only p% of examples were actually modified by Multi-VALUE then I think the theoretical headroom over an SAE baseline should be p% absolute accuracy, and probably much less in practice if the modifications were minor. Also, a low change rate may indicate that some aspect of the test sets (such as domain, topic, or possibly the task itself) doesn't have real-world value to native speakers: for example, if dialectal differences only show up in colloquial text, then there's little-to-no value testing on formal text.

Question E) In Table 2, how does HyperLoRA have fewer parameters than LoRA? My understanding was that your LoRA model uses a single set of matrices for all dialects, while your HyperLoRA model uses a a set of hypernetworks whose output shape is the same as the LoRA matrices. It's possible I've misunderstood the architecture, but it seems to me that you would need strictly more parameters for HyperLoRA.

Question F) Could you provide more detail about the feature representation for the expert dialectal knowledge? I looked at eWAVE and it appears to have 235 features, but each can take five values (that I saw). How do you encode these as input to the model, and for your coverage metric?

Question G) Table 5 seems to feature multiple dialects never mentioned in the text. It would be good to include their names somewhere and reword the Datasets portion of Section 4 to indicate that you don't exclusively use the 5 mentioned dialects.

Question H) It seems that the three rows in Table 5 are not the only combinations you tested; are there any useful summary statistics you could calculate over the whole set of experiments? For example, instead of only providing two anecdotal combinations, it would be interesting to see how well L1 distance and Coverage each correlate with downstream performance.

**Reasons To Accept:**

- Dialect data is generally scarce, so zero-shot methods like this one are widely applicable.
- Dialects are, in my opinion, under-researched.
- The baselines and ablations are thoughtfully constructed and thorough.
- The application of hypernetworks to LoRA appears to be novel.
- The paper acknowledges its own limitations and includes a thoughtful ethics statement, which should be encouraged.

**Reasons To Reject:**

- The test data is synthetic, generated by a rule-based system; we can't be sure that any of the results (baseline or experimental) accurately reflect performance on real data gathered from native speakers of these dialects. Lines 276-280 and the Limitations section discuss this, but in my opinion the discussion is inadequate. I would need to see citations and a specific, well-reasoned explanation for why it is valid to draw conclusions about real-world performance based on synthetic test data. This is my single largest issue with this paper.
- Many of the results appear to be disappointing; there is not a clear trend of HyperLoRA significantly improving over baseline performance.
- It is not clear that the test sets contain enough dialectal differences to estimate real-world benefit to native speakers.
- Some details about the method were unclear from the text (though I acknowledge that the promised public code release mitigates this somewhat).

**Reproducibility:**

4: Could mostly reproduce the results, but there may be some variation because of sample variance or minor variations in their interpretation of the protocol or method.

**Reviewer Confidence:**

4: Quite sure. I tried to check the important points carefully. It's unlikely, though conceivable, that I missed something that should affect my ratings.

**Typos Grammar Style And Presentation Improvements:**

In line 12 of the abstract, "prior work on dialect struggle" should either use "works" or "struggles".

In the equation on line 302, what are a, b, curly X, and curly Y don't appear to be defined anywhere.

The color-coding of the points in Figure 3 seem to repeat information encoded by their location on the x-axis. Consider removing the color-coding; also note that it will not be noticeable in black-and-white.

---

> ### Author Rebuttal · Authors · 2023-08-29
>
> Thank you for the effort you put in writing this detailed and thoughtful review. We really appreciate the questions and your interest in our work. Based on your feedback, we will revise the manuscript to better address your concerns on drawing conclusions about real-world performance from synthetic data and on the presence of dialectal differences in our test sets. First, we provide a detailed explanation and cite the authors of Multi-VALUE [1] on why it is reasonable to draw conclusions about modeling choices on real-world performance in the datasets section of our experimental setup. Secondly, we include additional statistics on the presence of dialectal differences in all test sets in the appendix.
>
> Your “single largest issue” highlights the gap in the current literature of a rich and diverse natural dataset to benchmark dialect robustness for English. While the use of pseudo-dialects can fail to exactly reflect real world performance, it does not impede on drawing conclusions about modeling choices. We see our paper as a first step towards providing some dialectal robustness and we encourage the community to improve the toolbox around dialectal variations. Whenever possible, we urge users to properly validate user-facing models with real-world in-domain data.
>
> **Question A: evidence that testing on synthetic data still allows us to draw conclusions about a hypothetical real-world scenario.**
>
> The use of synthetic data generated by Multi-VALUE to draw conclusions about modeling choices for dialectal performance is justified [1]. First of all, the authors of Multi-VALUE find that modifications to the model resulting in performance improvements on the synthetic data of Indian English and Chicano English also improve results on parallel gold standard natural dialect data provided by native speakers of Indian English and Chicano English. This implies that the results from testing on synthetic data are directionally correct, allowing us to draw conclusions about real-world performance.
>
> Furthermore, Multi-VALUE synthetic data has been validated by native speakers to be representative of their unique grammar constructions. (1) The dialect features found in Multi-VALUE are based on eWAVE, a database of observed morphosyntactic variations in 77 english varieties, compiled by a team of 84 professional linguists from descriptive materials, naturalistic corpus data, and native speaker knowledge [2]. This grounds Multi-VALUE in appropriate expert knowledge directly extracted from native speakers and naturalistic texts. (2) Multi-VALUE validated 92 of these rules by a group of 72 native speakers. With above 81% accuracy validated by native speakers and on average 86.6% implemented features per dialect, Multi-VALUE provides us with a reasonable level of reliability to draw conclusions on real-world dialect performance.
>
> We thank the reviewer again for pointing out this key concern and we will update the paper to better address this concern.
>
> [1] Caleb Ziems, William Held, Jingfeng Yang, Jwala Dhamala, Rahul Gupta, and Diyi Yang. 2023. Multi-VALUE: A Framework for Cross-Dialectal English NLP. ACL 2023.
>
> [2] Bernd Kortmann, Kerstin Lunkenheimer, and Katharina Ehret, editors. 2020. eWAVE.
>
> **Question B: The introduction mentions dialects without standardized writing systems as a specific use-case where zero-shot methods are especially beneficial.**
>
> Our approach does not address potential orthographic variations that arise from non-standard writing systems. As you point it out, these orthographic variations, much like lexical variations, give rise to additional difficulties in harnessing language technologies. To this effect, we revise this part of the introduction to focus on non-standard grammars. Our work is an important first step in quickly adapting to these new standards which vary in their grammatical rules.
>
> To answer your question, all dialects in our experiments have standard writing systems.
>
> **Question C: Your technique is clearly applicable in the setting where no labeled data is available, but what about when a small amount is?**
>
> This is an insightful remark, we thank you for the suggestion. At the moment, we focus on highlighting the use of expert knowledge encoded in dialect features, but we will be happy to extend the applicability of our work to small amounts of dialect data in the future.
>
> One way this can be done is by adding small amounts of data from many dialects to the HyperLoRa training set. We didn’t have time to add the experiment here, but the hypernetwork flexibly allows you to mix any amount of data from any number of dialects.
>
> **Question D: It is not clear that the test sets contain enough dialectal differences to estimate real-world benefit to native speakers.**
>
> In the table below, you can find the % of total entries in the test sets which have been modified by Multi-VALUE. On average, for each dialect, we have over 88% transformed entries except for Chicano English. This is expected, as Chicano English shares many similarities with Colloquial American English. In the case of Colloquial Singaporean English, the entries are almost always transformed by Multi-VALUE. It is difficult in practice to get a precise headroom estimate as dialect variations do not fit in deterministic baskets, instead different features are utilized at different rates. In our case, a large proportion of the test set is affected, and we expect theoretical headroom estimates to be large overestimates.
>
> | | CoLA (%) | MNLI (%) | QNLI (%) | RTE (%) | QQP (%) | SST2 (%) | STSB (%) | Mean (%) |
> | --- | -------- | -------- | -------- | -------- | -------- | -------- | -------- | -------- |
> | AAVE | 95.2 | 93.6 | 95.0 | 99.3 | 97.0 | 93.2 | 91.8 | 95.0 |
> | ChcE | 55.3 | 59.0 | 26.4 | 74.0 | 41.4 | 57.9 | 37.1 | 50.1 |
> | IndE | 98.8 | 96.8 | 99.6 | 100 | 98.8 | 97.1 | 99.6 | 98.7 |
> | NgE | 82.6 | 87.5 | 84.5 | 98.6 | 82.2 | 91.3 | 91.2 | 88.3 |
> | CollSgE | 99.7 | 97.6 | 99.5 | 100 | 99.7 | 97.1 | 99.8 | 99.1 |
>
> Furthermore, the applied features across the test set are diverse. In the following table, you can find the number of processed rules per dialect and task. Across all dialects, a large majority of rules are being applied to the test sets.
>
> | | Total Feat | CoLA | MNLI | QNLI | RTE | QQP | SST2 | STSB |
> | --- | -------- | -------- | -------- | -------- | -------- | -------- | -------- | -------- |
> | AAVE | 118 | 92 | 109 | 91 | 86 | 110 | 89 | 92 |
> | ChcE | 30 | 23 | 28 | 24 | 25 | 28 | 22 | 24 |
> | IndE | 90 | 71 | 85 | 77 | 68 | 84 | 74 | 73 |
> | NgE | 45 | 34 | 42 | 32| 35 | 42 | 34 | 32 |
> | CollSgE | 67 | 58 | 63 | 54 | 51 | 63 | 54 | 55 |
>
> Your comment highlighting the dependence of feature rates on the specific context is important. For instance, Nigerian pidgin is typically used in informal conversation, whereas Nigerian English is vastly used in politics, science, and media. Our work takes a first step towards task-agnostic and resource-efficient dialect adaptation, we think that further adaptation to topical and register shifts can be an interesting area of future research.
>
> **Question E: In Table 2, how does HyperLoRA have fewer parameters than LoRA?**
>
> Your understanding of HyperLoRA is correct, the total number of parameters in HyperLoRA is strictly greater than LoRA by the number of parameters of the hypernetwork. This is actually a typo from an earlier version where we wanted to report the number of trainable parameters. This will be removed in the revision to avoid confusion.
>
> **Question F: Could you provide more detail about the feature representation for the expert dialectal knowledge?**
>
> For each of those 235 features, eWAVE specifies different heuristic probabilities: 100% for obligatory features, 60% for features neither pervasive nor rare, 30% for rare features and 0% for no information or attested absence. These probabilities are used by MultiVALUE to generate synthetic dialect data. We directly use these probabilities to encode expert dialectal knowledge. For both the model and our coverage metric these feature vectors take values in {1, 0.6, 0.3, 0}^{# features}. We will revise the manuscript to include these details.
>
> **Question G: Table 5 seems to feature multiple dialects never mentioned in the text.**
>
> Thank you for catching this! We will carefully mention the dialects used in this ablation study in our dataset section in the revised manuscript.
>
> **Question H: are there any useful summary statistics you could calculate over the whole set of experiments?**
>
> We agree that it would be interesting to visualize potential correlation between downstream performance and the two metrics. However, such correlations risk being misleading given the relatively small sample of experiments we have at the moment. We will finalize these experiments by the end of the discussion period and adjust the appendix according to our findings.
>
> ** Other **
>
> Thank you for catching those issues with our grammar and presentation style! We have revised the paper to address these stylistic problems.
> - prior works on dialect struggle
> - a, b: these are typos (should be \alpha, \beta respectively)
> - \mathcal{X}, \mathcal{Y}: feature spaces for last layer dialect and SAE representations respectively
> - Following your comment, we have removed the color-coding and simply mention that “The cost of training scales linearly with the number of annotated samples”
>
> In addressing these questions, we hope that we were able to bring clarity to important issues you had. We would be happy to answer any additional concerns and we greatly appreciate any further feedback.

---

### Official Review · Reviewer_g9kY · 2023-08-12

**Soundness:** 4

**Excitement:**

4: Strong: This paper deepens the understanding of some phenomenon or lowers the barriers to an existing research direction.

**Paper Topic And Main Contributions:**

This paper proposes HyperLoRA, which combines the hypernetwork for the dialect topology and low-rank adaptation (LoRA) for efficient fine-tuning, to implement efficient dialect adaptation with improved generalization and good results on zero-shot dialect experiments.

**Reasons To Accept:**

1. Novel to combine hypernetwork with LoRA for the domain adaptation
2. Very clear writing, demonstration, and math expression
3. The results demonstrate HyperLoRA can be extendable (works on unseen dialects), and very efficient (w/o slowing inference)

**Reasons To Reject:**

1. The paper only works on RoBERTa-Base model and lack of the proof of scalability.

**Reproducibility:**

4: Could mostly reproduce the results, but there may be some variation because of sample variance or minor variations in their interpretation of the protocol or method.

**Reviewer Confidence:**

2: Willing to defend my evaluation, but it is fairly likely that I missed some details, didn't understand some central points, or can't be sure about the novelty of the work.

---

> ### Author Rebuttal · Authors · 2023-08-29
>
> Thank you for your thoughtful review. Your concern on the applicability of this method to models other than RoBERTa-Base is important to us. Experiments for HyperLoRA are currently limited to encoder-only models and do not account for encoder decoder, or decoder-only models. We did not find the time to complete these experiments. We leave the evaluation of HyperLoRA and the development of dialect adaptation methods on these other architectures as future work.

---

### Meta-Review · Area_Chair_Qd3M · 2023-09-19

**Recommendation:** 3

**Metareview:**

This paper explores how to use linguistic knowledge to enable resource-efficient adaptation via hyper-networks on 5 English dialects. The most interesting part of this paper is the usage of expert knowledge instead of labeled data. The paper sheds light on an interesting and under-studied phenomena: dialects. In general the reviewers agree that the proposed method is strong, as well as the experimental setup and the used baselines. Additionally ablation studies and a in-depth analysis was also performed.

The limitations of this paper are: that the method is only tested on RoBERTa-base (g9kY);  that it is tested on synthetic data and that this dataset is not well describe and might be insufficient to measure the linguistical diversity (i59w) ; and a lack of proper comparison with other dialectic adaptation methods (wWbU).

Overall the reviewers have strong scores for this paper. However, I would also take in consideration the weaknesses pointed out.

---

### Decision · Program_Chairs · 2023-10-07

**Decision:**

Accept-Main

**Comment:**

This paper explores how to use linguistic knowledge to enable resource-efficient adaptation via hyper-networks on 5 English dialects. The most interesting part of this paper is the usage of expert knowledge instead of labeled data. The paper sheds light on an interesting and under-studied phenomena: dialects. In general the reviewers agree that the proposed method is strong, as well as the experimental setup and the used baselines. Additionally ablation studies and a in-depth analysis was also performed.

The limitations of this paper are: that the method is only tested on RoBERTa-base (g9kY);  that it is tested on synthetic data and that this dataset is not well describe and might be insufficient to measure the linguistical diversity (i59w) ; and a lack of proper comparison with other dialectic adaptation methods (wWbU).

Overall the reviewers have strong scores for this paper. However, I would also take in consideration the weaknesses pointed out.